# Design and Control of the Manipulator of Magnetic Surgical Forceps with Cable Transmission

**DOI:** 10.3390/mi16060650

**Published:** 2025-05-29

**Authors:** Jingwu Li, Zhijun Sun

**Affiliations:** State Key Laboratory of Mechanics and Control of Mechanical Structures, Nanjing University of Aeronautics and Astronautics, Nanjing 210016, China; lijingwu@nuaa.edu.cn

**Keywords:** magnetic and cable-driven surgical forceps, manipulator, decoupling control

## Abstract

Magnetically actuated medical robots have attracted growing research interest because magnetic force can transmit power in a non-contact manner to fix magnetic surgical instruments onto the inner wall of the abdominal cavity. In this paper, we present magnetic and cable-driven surgical forceps with cable transmission. The design achieves significant diameter reduction in the manipulator by separating the power sources (micro-motors) from the manipulator through cable transmission, consequently improving surgical maneuverability. The manipulator adopting cable transmission mechanism has the problem of joint motion coupling. Additionally, due to the compact space within the magnetic surgical forceps, it is difficult to install pre-tightening or decoupling mechanisms. To address these technical challenges, we designed a pair of miniature pre-tensioning buckles for connecting and pre-tensioning the driving cables. A mathematical model was established to characterize the length changes of the coupled joint-driving cables with the angles of moving joints and was integrated into the control program of the manipulator. Joint motion decoupling was achieved through real-time compensation of the length changes of the coupled joint-driving cables. The decoupling and control effects of the manipulator have been verified experimentally. While one joint moves, the angle changes of the coupled joints are within 2°.

## 1. Introduction

Compared with open surgery, minimally invasive surgery (MIS) significantly reduces surgical trauma to the human body, with a better recovery effect [1,2,3,4]. While performing minimally invasive surgery, each surgical instrument is introduced into the abdominal cavity through a dedicated hole, and surgical instruments cooperate with each other to perform surgical operations. Single-port surgical robots only need to cut an incision to introduce all surgical instruments, which further minimizes the damage caused by surgical procedures [5]. Since all surgical instruments of the single-port surgical robot are inserted into the abdominal cavity through a single incision, there is a problem of collision and interference between the surgical instruments. [6].

Surgical instruments with a magnetic anchoring and guidance system (MAGS) [7] can be introduced into the abdominal cavity through one incision. There are several built-in permanent magnets, called internal permanent magnets (IPMs), in the body of the magnetic surgical instrument. The magnetic surgical instrument will be anchored onto the inner wall of the abdominal cavity through magnetic coupling between the embedded IPMs and external permanent magnets (EPMs) positioned extracorporeally. Magnetic surgical instruments can move freely along the inner wall of the abdominal cavity without having the abdominal fulcrum constrain., and because magnetic surgical instruments are separated from each other, there is no interference between them.

In addition to anchoring and positioning magnetic surgical instruments, magnetic force can also be used to drive surgical operations. The magnetic surgical instruments equipped with IPMs are actuated by the external magnetic field, which can be precisely controlled through current regulation in multiple extracorporeal electromagnets [8,9]. In some designs, the EPMs can drive magnetic surgical instruments to perform certain surgical maneuvers [10,11,12,13].

Motors provide an alternative actuation method for magnetic surgical instruments, enabling various motions, including telescoping [14], deflection [15], opening and closing [16], and clamping [17,18]. Due to the presence of motors, the diameter of the magnetic surgical instrument’s manipulator tends to be large, which is not conducive to the cooperation between magnetic surgical instruments. Additionally, the surgeon’s visual field is more obstructed by magnetic surgical instruments, making it inconvenient to observe surgical procedures.

The manipulators of minimally invasive surgical robots adopt a continuum structure [19,20], cable-driven structure [21,22], or other structures [23,24,25,26], and the power sources of these structures are separated from the manipulators. Thanks to the simple structure, the manipulator has a smaller diameter, which is more suitable for surgical operations. Li designed a magnetic anchored and cable-driven endoscope with a cable-driven mechanism in the internal unit [27]. This mechanism employed two pairs of cables to drive the tilting motion along two directions. But the cables may impose limitations on the endoscope’s orientation. The magnetic actuator developed by Valdastri [28] has a continuum structure. Two built-in magnets act as rotors to drive the manipulator’s movement. This design still has technical challenges to be solved about reducing the volume and weakening the magnetic interference.

Similar to the minimally invasive surgical robots with cable-driven structures, we designed a magnetic surgical forceps characterized by using cables to transmit power. The magnetic surgical forceps employ built-in motors within its body as the power source. Only the wires for transmitting current or data are connected to the device outside of the abdominal cavity via the incision, which have no influence on the movement of the magnetic surgical forceps.

The cable-driven manipulator of the proposed magnetic surgical forceps has the problem of joint motion coupling. A decoupling mechanism is a good method to realize the precise control of the manipulator [29,30,31,32,33,34,35], whereas it will increase the complexity and diameter of the manipulator. This is not suitable for application in magnetic surgical instruments that have stringent requirements for small dimensions. When a certain joint moves, the length of the coupled joint-driving cables will change, inducing coupling motion. In this work, we achieve the decoupling by actively controlling motors to compensate for the length changes of the coupled joint-driving cables. This method requires no additional mechanisms, enabling decoupling purely through control algorithms, without increasing the size of the magnetic surgical instrument.

Manipulators with a small diameter are beneficial for the mutual cooperation between surgical instruments to complete surgical operations. Additionally, their smaller size reduces obstruction of the surgical field, improving the surgeon’s visibility during procedures. The i^2^Snake robotic platform designed by Yang [36,37] features a manipulator with a diameter of only 3–4 mm. Compared to traditional minimally invasive surgical robots, all components of the magnetic surgical instruments are integrated within its body. This design constraint makes miniaturizing the manipulator diameter more challenging. The modular magnetic platform developed by Tortor [38,39] integrates several surgical instruments whose manipulators provide 4 DoFs with a diameter of 12 mm. To maintain this compact design, the motors integrated into these instruments are only 4 mm in diameter. The magnetic anchored miniature robot designed by Oleynikov [40,41] has two manipulators, each having 4 DoFs with a diameter of 26 mm. Feng [14] designed a magnetic anchored surgical grasper with a length of 110 mm; the manipulator has 3 DoFs for deflection, extension, and clamping, with a diameter of 20 mm. Compared to other magnetic anchored surgical instruments (not including magnetic anchored laparoscope), the magnetic surgical forceps we designed utilizes a cable-driven mechanism to achieve the smallest diameter of the manipulator.

The remainder of this paper is organized as follows. Section 2 illustrates the structure of the magnetic surgical forceps and details the pre-tensioning buckle and anti-sliding design for cable transmission. A kinematic analysis of the manipulator is carried out in Section 3. Section 4 explains the length variations of the coupled joint-driving cables with the angles of moving joints. In Section 5, the decoupling effect was verified experimentally, and a grasping experiment was conducted to preliminarily verify the working ability of the magnetic surgical forceps.

## 2. Structure and Manufacturing of the Magnetic Surgical Forceps

Magnetic surgical forceps comprise two parts: a magnetic anchored and steered part and a manipulator, as shown in Figure 1. Two ring-shaped IPMs (inner diameter: 16 mm, outer diameter: 22 mm, and thickness: 10 mm) were mounted at two ends of the magnetic anchored and steered part to generate magnetic force with the EPMs. A magnetic anchoring device consists of two EPMs, as shown in Figure 2. Once the magnetic surgical forceps are inserted into the abdominal cavity, they are attached to the inner wall of the abdominal cavity by the magnetic anchoring device. While the magnetic anchoring device is repositioned or rotated along the outer wall of the abdominal cavity, the magnetic surgical forceps follow the motion along the inner wall of the abdominal cavity.

The motors installed in the magnetic anchored and steered part serve as the power source to drive the motion of the manipulator. Cables are used to connect the motors and the manipulator joints. A pair of bevel gears mounted perpendicularly are used to change the orientation by 90°. The bevel gears have grooves for cable winding, and cables are connected to the joints via guiding holes and pulleys.

The pulleys are used to alter the path of the driving cables at the pitch joint. Friction occurs between the cables and shafts during actuation. The static friction between the driving cable and pulley exceeds that between the pulley and the shaft. Consequently, relative sliding occurs at the interface between the pulley and shaft during operation. The effect of the friction between the driving cable and pulley is to offset the friction resistance between the pulley and the shaft.

Cable-driven transmission is commonly employed in minimally invasive surgical robots [2,42], but it is rarely adopted in magnetic surgical instruments. In this paper, for the first time, the magnetic surgical instrument utilizes a cable transmission mechanism to transmit power or motion from the micro-motors to the manipulator. The micro-motors are separated from the manipulator through cables, thus reducing the manipulator’s diameter. We achieved a significantly reduced manipulator diameter of 10 mm, which is close to the conventional minimally invasive robots and smaller than developed magnetic surgical instruments (not including magnetic endoscope).

The joints are actuated through friction between the cables and the joint shafts. Excessive cable pre-tension makes it difficult to actuate the joints, while insufficient preload results in significant slippage between the cables and the joint shafts, so adjusting the preload force to a proper level is important. The magnetic surgical forceps are small for operating in the abdominal cavity, making it challenging to connect two ends of each cable and pre-tension the cables. To address this, a pair of pre-tensioning buckles were designed and manufactured to fasten and preload the driving cables, as shown in Figure 3. Each end of the cables was tied to a pre-tensioning buckle; pre-tensioning buckle 1 can be screwed into pre-tensioning buckle 2 up to 5 mm to make the driving cables tighter. The pre-tightening of the driving cables can be regulated by adjusting the insertion depth (within 5 mm) of the pre-tensioning buckle 1 into pre-tensioning buckle 2.

Due to the constrained assembly space within the shell of the magnetic surgical forceps, it is difficult for human hands or other tools, such as tweezers, to perform the rotational connection between the two pre-tensioning buckles. So, four holes with a diameter of 0.5 mm are uniformly and circumferentially distributed in the pre-tensioning buckle 1 to insert steel rods to assist in connecting the two pre-tensioning buckles, as shown in Figure 3.

To minimize relative sliding between the bevel gears and the cables, as well as between the joint shafts and the cables, holes are set in the bevel gears and joint shafts for the cables to pass through. The cable will not only wrap around the bevel gear and joint shaft but will also pass through the holes with an effect similar to tying a knot at the holes, as illustrated in Figure 4. Experimental results demonstrate no observable relative sliding between the bevel gears and cables, nor between the cables and the joint shafts during manipulator operation.

The prototype of the magnetic surgical forceps and their related components are shown in Figure 5. Most components, including the housing and manipulator, were fabricated using 3D printing with a layer thickness of 0.075 mm. The bevel gears and the plate with guiding holes were machined from pure copper with 0.02 mm precision. The supporting rod was made of stainless steel with a smooth surface (Ra = 0.8 µm). The magnetic anchored unit has a diameter of 25 mm and a length of 110 mm. Clinical trials suggest that a length of 110 mm for magnetic surgical instruments attached to the inner wall of the abdominal cavity is practical [43]. The motors used have a diameter of 8 mm, a length of 34 mm, and their output torque is 50 mN · m. Each motor independently drives one joint of the manipulator. The shoulder joint enables pitch motion, the elbow joint enables yaw motion, and the surgical forceps head performs the grasping function.

## 3. Kinematic Analysis of the Manipulator

The manipulator of the magnetic surgical forceps has three DoFs, which are pitch, yaw, and clamping in sequence. The specific structure of the manipulator is depicted in Figure 6. Each driving cable is wound around a joint shift to drive motion. The DoFs of pitch and yaw are responsible for controlling the position and orientation of the end-effector of the magnetic surgical forceps. Considering that the manipulator of the magnetic surgical forceps is serially connected by three joints, the Denavit–Hartenberg (D–H) method was used to calculate the forward kinematics of the manipulator. Three local coordinate systems $\left\{E_{i}\right\}\left(i=1,2,3\right)$ corresponding to three joints respectively and a base coordinate system $\left\{E_{0}\right\}$ are set as shown in Figure 6a.

The position of the manipulator’s tip is(1)P=T10·T21·T32·P3
where *P* represents the position of the manipulator’s tip, ^3^*P* represents the position vector of the manipulator’s tip in the local coordinate system $\left\{E_{3}\right\}$, and Tnn−1(n=1,2,3) is the transformation matrix from the local coordinate system $\left\{E_{i}\right\}$ to $\left\{E_{i-1}\right\}$. The transformation matrices between the local coordinate systems are(2)T10=cos(u)−sin(u)00sin(u)cos(u)0000100001T21=cos(v)−sin(v)05000−10sin(v)cos(v)000001T32=10030010000100001

The pitch, yaw, and clamping joints have lengths of 50 mm, 30 mm, and 20 mm, respectively. Due to structural constraints, the maximum bending angle of the pitch joint is limited to 70°. The magnetic anchored and steered part has a diameter of 25 mm. The centerline of the rotation shaft of the pitch joint intersects with the axis of the magnetic anchored and steered part. Thus, the deepest position that the manipulator’s tip can reach is 106.45 mm away from the inner wall of the abdominal cavity. The motion range of the yaw joint is from −40° to 40°. The workspace is shown in Figure 7.

## 4. The Coupling Among Manipulator Joints

### 4.1. Motion Coupling of the Manipulator Joints

The manipulator has three DoFs: pitch, yaw, and clamping. These three DoFs are actuated by three driving cables respectively, noted as $L_{1}$, $L_{2}$, $L_{3}$. To characterize the length changes of these three driving cables during joint motion, the driving cables are divided into multiple sections according to the location, as illustrated in Figure 8. Regarding the coupling motion, only certain sections of the coupled joint-driving cables undergo length changes. Therefore, by analyzing the length variations of these cable sections, the total change can be determined.

$L_{211}$ represents a section of the driving cable $L_{2}$, which is from the pulley’s apex to the tangent point between $L_{21}$ and the yaw joint shaft. $L_{221}$ is defined similarly to $L_{211}$, denoting the cable section from the pulley’s apex to the tangent point between $L_{22}$ and the yaw joint shaft. $L_{311}$ and $L_{321}$ are defined in the same way as $L_{211}$ or $L_{221}$. During the pitch joint motion, the lengths of sections $L_{211}$, $L_{221}$, $L_{311}$, and $L_{321}$ of the driving cables used for driving the coupled joints will change, and when the yaw joint of the manipulator moves an angle, the length of sections $L_{311}$ and $L_{321}$ of the clamping joint-driving cable has variations.

In this section, we explain the relationship between the length variations of the coupled joint-driving cables and the moving joints’ rotation angles through structural analysis. Decoupling can be achieved by compensating for the length changes in these coupled joint-driving cables.

### 4.2. The Yaw Joint Coupling with the Pitch Joint

While the pitch joint of the manipulator is moving, the lengths of $L_{211}$ and $L_{221}$ change. The difference of $L_{211}-L_{221}$ drives the coupling motion in the yaw joint. A sketch of the yaw joint-driving cable sections that pass through the manipulator is shown in Figure 9. For the convenience of observation and calculation, the top view is chosen to be perpendicular to the manipulator, as shown in the bottom left of Figure 9b. The values of parameters in Figure 9 are listed in Table 1.

$L_{211}$ and $L_{221}$ are(3)$$\begin{array}{cc} \; & L_{211}=O_{21}+l_{211}\\ \; & L_{221}=O_{22}+l_{221} \end{array}$$

$O_{21}$ represents the overlapping part between $L_{211}$ and the pulley, $O_{22}$ represents the overlapping part between $L_{221}$ and the pulley, and $l_{211}$ and $l_{221}$ are the straight-line portions of $L_{211}$ and $L_{221}$, respectively. In the initial stage, the manipulator is parallel to the axis of the magnetic anchored and steered part with the pitch joint angle at zero.

$l_{211}$ and $l_{221}$ are(4)$$\begin{array}{cc} \; & l_{211}=\sqrt{d_{1}^{2}+d_{2}^{2}-r_{p}^{2}+{\left(r_{s}+d_{6}\right)}^{2}}\\ \; & l_{221}=\sqrt{d_{5}^{2}-r_{p}^{2}+{\left(r_{s}-d_{6}\right)}^{2}} \end{array}$$

In (Equation 4),(5)$$d_{5}=\sqrt{{\left(P_{1x}-C_{2x}\right)}^{2}+{\left(P_{1y}-C_{2y}\right)}^{2}}$$

$P_{1}$ represents the projections of two vertices on two sides of the contact portion between the yaw joint shaft and the driving cable on the front plane. The projections of these two vertices are both located on the centerline of the yaw joint shaft. $P_{1x}$ and $P_{1y}$ are $P_{1}$ horizontal and vertical coordinates. $C_{2x}$ and $C_{2y}$ are coordinate values of the central point of the pulley $C_{2}$.

$O_{21}$ and $O_{22}$ are(6)$$\begin{array}{cc} O_{21} & =r_{p}*\left(\chi +\alpha +\beta \right)\\ \; & =r_{p}*\left(\chi +\sin^{-1}\left(\frac{r_{p}}{d_{4}}\right)+\tan^{-1}\left(\frac{d_{2}}{d_{1}}\right)\right) \end{array}$$(7)$$\begin{array}{cc} O_{22} & =r_{p}*\left(\varepsilon +\phi \right)\\ \; & =r_{p}*\left(\sin^{-1}\left(r_{s}/d_{5}\right)+\tan^{-1}\left(\frac{C_{2y}-P_{1y}}{P_{1x}-C_{2x}}\right)\right) \end{array}$$

### 4.3. The Clamping Joint Coupling with the Pitch Joint and the Yaw Joint

The clamping joint-driving cable passes through the holes in the yaw joint to the clamping joint. The holes have a four-pronged conical shape. Figure 10 illustrates the structure and dimensions of the holes. The contact states between the clamping joint-driving cable and the two holes are depicted in Figure 11. $\gamma_{1}$, $\gamma_{2}$, $\gamma_{3}$, and $\gamma_{4}$ are the critical angles governing contact state transitions. These angles can be determined by calculating the yaw joint angle when the clamping joint-driving cable just makes contact with one sidewall of the two holes. When the deflection angle of the yaw joint is in the range of $\gamma_{1}\leq \gamma <\gamma_{2}$, the cable section $L_{311}$ will have an overlapping part with a lateral surface of a hole. When the deflection angle of the yaw joint is larger than $\gamma_{2}$, both $L_{311}$ and $L_{321}$ contact the lateral surfaces of their respective holes. Similarly, for deflection angle in the range $\gamma_{4}<\gamma \leq \gamma_{3}$, only cable section $L_{321}$ contacts a hole’s lateral surface. When the angle is smaller than $\gamma_{4}$, both $L_{311}$ and $L_{321}$ contact the lateral surfaces of both holes.

Figure 12 demonstrates the clamping joint-driving cable in all six phases, with corresponding parameters listed in Table 2. Here, $P_{1}$ denotes the center point of the hole traversed by $L_{311}$, while $P_{3}$ represents the center point of the hole traversed by $L_{321}$. $P_{3}$ denotes the intersection between $L_{311}$ and the hole’s sidewall, related to the state $\gamma_{4}<\gamma \leq \gamma_{3}$ in Figure 11. $P_{4}$ denotes the intersection between $L_{321}$ and the hole’s sidewall for the state $\gamma_{1}\leq \gamma <\gamma_{2}$. β and γ represent the deflection angles of the pitch joint and yaw joint, respectively.

The length change of the clamping joint-driving cable, caused by the coupling with either the pitch joint or yaw joint, occurs exclusively at the location preceding the holes’ central points in the yaw joint. Specifically, these length changes are confined to cable sections $L_{311}$ and $L_{321}$. In phase 1 ($0\leq \gamma <\gamma_{1}$), there is no touch between the clamping joint-driving cable and the holes’ sidewalls.(8)$$L_{321}=l_{321}+O_{32}$$

In this configuration, $L_{3111}$ represents the frontal projection of the straight-line portion of $L_{311}$.(9)$$L_{3111}=\sqrt{{\left(d_{1}-d_{11}/2*\sin\left(\gamma \right)\right)}^{2}-r_{p}^{2}}$$

$l_{311}$ denotes the straight-line portion of $L_{311}$,(10)$$l_{311}=\sqrt{L_{3111}^{2}+{\left(d_{11}/2*\cos\left(\gamma \right)-d_{9}\right)}^{2}}$$

$O_{31}$ denotes the overlapping part between $L_{311}$ and the pulley,(11)$$\begin{array}{c} O_{31}=r_{p}*\chi =r_{p}*\left(\alpha +\beta \right) \end{array}$$
where α represents the angle between $L_{311}$ and the central axis of the pitch joint. The value of α can be obtained by(12)α=sin−1(rp/P1x2E1+P1y2E1)
where P1xE1 and P1yE1 denote the x- and y-coordinates of $P_{1}$ in the coordinate system $\left\{E_{1}\right\}$.(13)P1xE1=(d1−d11/2∗sin(γ))∗cos(β)P1yE1=−(d1−d11/2∗sin(γ))∗sin(β)

It can be known that(14)$$L_{311}=l_{311}+O_{31}$$

Similarly, $l_{321}$ denotes the straight-line portion of $L_{321}$, and $O_{32}$ represents the overlapping part of $L_{321}$ and the pulley. $L_{321}$ is(15)$$L_{321}=l_{321}+O_{32}$$

When $L_{321}$ aligns parallel to the axis of the magnetic anchored and steered part, the deflection angle of the pitch joint is $\beta_{1}$, resulting in $O_{32}=0$. Consequently, for any pitch joint deflection angle smaller than $\beta_{1}$, the value of $O_{32}$ remains identically zero.(16)$$O_{32}=\left\{\begin{array}{cc} 0 & \beta \leq \beta_{1}\\ r_{p}*\left(\delta -\epsilon \right) & \beta >\beta_{1} \end{array}\right.$$

$\beta_{1}$ can be calculated by(17)$$\beta_{1}=\sin^{-1}\left(\frac{d_{2}}{d_{1}+d_{11}/2*sin\left(\gamma \right)}\right)$$
in (Equation 16),(18)$$\begin{array}{cc} \; & \delta =\sin^{-1}\left(\frac{r_{p}}{\sqrt{{{(}^{E1}P_{2x}-J_{3x})}^{2}+{{(}^{E1}P_{2y}-J_{3y})}^{2}}}\right)\\ \; & \epsilon =\tan^{-1}\left({(}^{E1}P_{2y}-J_{2y})-{(}^{E1}P_{2x}-J_{2x})\right) \end{array}$$

P2xE1 and P2yE1 are coordinate values of $P_{2}$ in the coordinate system $\left\{E_{1}\right\}$. $J_{3}$ is the apex of the pulley $C_{2}$. $J_{2}$ is the intersection between $L_{321}$ and the pulley. P2xE1 and P2yE1 is(19)P2xE1=(d1+d112∗sin(γ))∗cos(β)P2yE1=−(d1+d112∗sin(γ))∗sin(β)
and(20)$$\begin{array}{c} J_{2x}==\left\{\begin{array}{cc} -d_{3} & \beta \leq \beta_{1}\\ -d_{3}+r_{p}*\sin\left(\delta -\epsilon \right) & \beta >\beta_{1} \end{array}\right.\\ J_{2y}==\left\{\begin{array}{cc} -d_{2} & \beta \leq \beta_{1}\\ -d_{2}-r_{p}*\left(1-\cos\left(\delta -\epsilon \right)\right) & \beta >\beta_{1} \end{array}\right. \end{array}$$

$L_{3211}$ represents the projection of the straight-line portion of $L_{321}$ onto the x-y plane of the coordinate system $\left\{E_{1}\right\}$.(21)$$L_{3211}=\left\{\begin{array}{cc} \sqrt{{{(}^{E1}P_{2x}-J_{3x})}^{2}+{{(}^{E1}P_{2y}-J_{3y})}^{2}} & \beta \leq \beta_{1}\\ \sqrt{{{(}^{E1}P_{2x}-J_{3x})}^{2}+{{(}^{E1}P_{2y}-J_{3y})}^{2}-r_{p}^{2}} & \beta >\beta_{1} \end{array}\right.$$

Then, the straight-line portion of $L_{321}$ is(22)$$l_{321}=\sqrt{L_{3211}^{2}+{\left(d_{11}/2*\cos\left(\gamma \right)+d_{8}\right)}^{2}}$$

So, the length of $L_{321}$ is(23)$$L_{321}=\left\{\begin{array}{cc} \sqrt{{{(}^{E1}P_{2x}-J_{3x})}^{2}+{{(}^{E1}P_{2y}-J_{3y})}^{2}} & \beta \leq \beta_{1}\\ \sqrt{{{(}^{E1}P_{2x}-J_{3x})}^{2}+{{(}^{E1}P_{2y}-J_{3y})}^{2}-r_{p}^{2}}+r_{p}*\left(\delta -\epsilon \right) & \beta >\beta_{1} \end{array}\right.$$

The above is the calculation process of $L_{311}$ and $L_{321}$ in phase 1 ($0\leq \gamma <\gamma_{1}$). The changes of $L_{311}$ and $L_{321}$ reflect the change of the clamping joint-driving cable. The lengths of $L_{311}$ and $L_{321}$ in phase 2 ($\gamma_{1}\leq \gamma <\gamma_{2}$) can be obtained by(24)$$\begin{array}{c} L_{311}=\sqrt{{\left(d_{8}-\frac{1}{2}\left(d_{11}\right)\cos\left(\gamma \right)\right)}^{2}+{\left(d_{1}-\frac{1}{2}d_{11}\sin\left(\gamma \right)\right)}^{2}-r_{p}^{2})}\\ +r_{p}*\left(\frac{r_{p}}{d_{1}-\sqrt{d_{16}^{2}+{\left(d_{17}+\frac{d_{11}}{2}\right)}^{2}}}+\beta \right)+\sqrt{d_{16}^{2}+d_{17}^{2}} \end{array}$$(25)$$\begin{array}{c} L_{321}=\left\{\begin{array}{c} \sqrt{{\left(\left(d_{1}-d_{18}\cos\left(\gamma +\iota \right)\right)*\cos\beta +d_{3}\right)}^{2}+{\left(\left(d_{18}\cos\left(\gamma +\iota \right)-d_{1}\right)*\sin\beta +d_{2}\right)}^{2}}\;\;\;\beta \leq \beta_{1}\\ \begin{array}{c} \sqrt{{\left(\left(d_{1}-d_{18}\cos\left(\gamma +\iota \right)\right)*\cos\beta +d_{3}\right)}^{2}+{\left(\left(d_{18}\cos\left(\gamma +\iota \right)-d_{1}\right)*\sin\beta +d_{2}+r_{p}\right)}^{2}-r_{p}^{2}}\\ +r_{p}*\left(\delta -\epsilon \right)\;\;\;\beta >\beta_{1} \end{array} \end{array}\right.\\ d_{18}=\sqrt{d_{16}^{2}+{\left(\frac{d_{11}}{2}-d_{17}\right)}^{2}} \end{array}$$

This section derives the length changes of the clamping joint-driving cable in phase 1 and phase 2. The length changes under other states described in Figure 12 are similar to phase 1 or phase 2, and thus are not detailed here.

## 5. Experimental Validation

### 5.1. Experiment Platform Setup

A motion capture system (NOKOV) with six cameras was used to record the joint motions, as shown in Figure 13. After calibration, the recognition precision of the motion capture system is 0.15 mm. The manipulator was controlled by a motor microcontroller (based on STM32F407), specifically designed for stepper motor control. Three stepper motors (ZWPD008008-369) used to drive three joints of the manipulator respectively were connected to the microcontroller. Three stepper motor drivers (TB6600) were used to send electrical signals to the stepper motors to drive their motion.

The motion commands are sent from the computer to the microcontroller through a serial port. The microcontroller receives commands, calculates the required pulse counts for each motor, and then transmits the motor driving signals to the motor drivers. Three motor drivers drive three stepper motors respectively according to the signals from the microcontrollers, thus driving the motion of the manipulator.

### 5.2. Joint Decoupling

Section 3 explains the relationship between the length changes of the coupled joint-driving cables and the angles of the moving joints. This model was integrated into the control program installed on the microcontroller. When a joint moves an angle, the microcontroller calculates the length changes of the driving cables connected to the coupled joints, and then controls the stepper motors used for driving the coupled joints to rotate to compensate for the length changes, so as to keep the coupled joints stable.

To evaluate the decoupling performance, when one joint is actuated, the motion capture system records the angle variations of the coupled joints. Due to structural constraints, the maximum angle of the pitch joint is 70°. In the experiment, when the pitch joint was commanded to move to 70°, the pitch joint always reached the maximum angle owing to control error, which cannot reflect the control accuracy at this angle. Consequently, we controlled the pitch joint to rotate from 0° to 60° with a step of 10°, and the motion capture system recorded the angles of the yaw joint and clamping joint, as shown in Figure 14b,c. The angles of the yaw joint and clamping joint exhibit slight fluctuations during operation. These fluctuations arise from two primary sources: (1) inherent mechanical vibration of the manipulator during motion; (2) measurement noise in the motion capture system during signal acquisition. To better reflect the changing trend of the joint angle, the data are smoothed through the adjacent average method.

The yaw joint initially measured 4.38°, while the clamping joint measured 2.65°. While the pitch joint rotated from 0° to 60°, the yaw joint angle exhibited a slight increase, whereas the clamping joint angle showed a minor decrease. As the pitch joint was driven to 60°, the angle of the yaw joint was 6.28°, and that of the clamping joint was 1.92°. Through five experimental repetitions, the smoothed data revealed angular variations of approximately 2° for the yaw joint and less than 1° for the clamping joint.

During operations, when the angle of the yaw joint exceeds |40°|, the joint’s control accuracy degrades significantly. When the yaw joint was in phase 3 or phase 6, the friction between the clamping joint-driving cable and the holes in the yaw joint got worse with the increase of the yaw joint deflection angle. Consequently, the motion of the yaw joint is limited in the range of −40° to 40°. During experiments, the yaw joint rotated from −40° to 40° with a step of 10°, as shown in Figure 15. The clamping joint remains basically stable with a variation of the smoothed data within 1°.

The experimental results demonstrated the feasibility of the cable length compensation method for motion decoupling. Within the range of joint motions, the coupled joints maintained stability, with angular variation below 2°. Another reason for the stability of the coupled joints is the hysteresis of driving cables. As long as the force from the driving cables is less than the inertia force of the coupled joints, the coupled joints have no motion. This inherent characteristic further contributes to the stability of the coupled joints.

### 5.3. Control Accuracy of the Manipulator

Experiments were carried out to verify the control accuracy of the three joints of the manipulator. A tendon-sheath mechanism has been widely used in minimally invasive surgical robots [44]. For the tendon-sheath mechanism, models [37,45,46,47] or mechanical structures [48] are often required to compensate for the cable backlash and hysteresis effects in order to achieve precise control. The manipulator of the magnetic surgical forceps adopts a cable pulley system, which exhibits lower friction losses with the contacting parts compared to the tendon sheath mechanism. As a result, the effects of backlash and hysteresis are weaker. Unlike minimally invasive surgical robots, where the manipulator is inserted into the abdominal cavity through a dedicated incision, the cable transmission distance in the magnetic surgical forceps’ manipulator is shorter. This further attenuates the backlash and hysteresis effects. During the initial startup stage of the magnetic surgical forceps’ manipulator driving experiments, obvious angular errors occur in all three joints due to the backlash and hysteresis effects. We assessed the hysteresis-induced angular errors for each joint during startup based on the results of multiple experiments. The control program of the manipulator compensates for the angular errors when the joints are initially actuated.

The control experiment of the manipulator joints is shown in Figure 16. Each set of experiments was repeated five times. The motion range of the pitch joint was from 0° to 60°. When the joint was larger than 10°, the motion angle of the pitch joint was slightly greater than the command. The average error is 1.51°, and the maximum error is 2.19° at 60°. The yaw joint rotated from −40° to 40°, with a step of 10°. The average angle error is 1.32°, with a maximum angle error of 2.33° at 0°. The motion range of the clamping joint was from 0° to 120°, with an increment of 20°. The average error is 3.48°. The motion angle of the clamping joint was slightly larger than the command.

### 5.4. Grasping Experiment

To simulate the operation of grasping human tissues during surgical procedures, we conducted an object grasping experiment to preliminary verify the working ability of the magnetic surgical forceps. A magnetic anchoring device comprising two EPMs used for generating attracting force with the IPMs embedded in the magnetic surgical forceps was placed on an acrylic plate with a thickness of 15 mm. The magnetic surgical forceps were attached to the lower surface of the acrylic plate by magnetic force, as shown in Figure 17a. The magnetic surgical forceps follow the position of the magnetic anchoring device. This grasping and releasing experiment also demonstrates independent control of the three joints of the manipulator.

First, the pitch joint was driven to orient the surgical forceps head at the end of the manipulator toward the yellow gripping object, as shown in Figure 17b. Then, the magnetic surgical forceps were moved to approach the gripping object by repositioning the magnet anchoring device. After the gripping object was picked up by the surgical forceps head, the pitch joint was driven to lift the manipulator. Next, the magnetic surgical forceps were repositioned by following the translation of the magnetic anchoring device, as shown in Figure 17f. Finally, the object was placed on the surface of the acrylic plate.

## 6. Conclusions

Cable transmission is a common mechanical structure adopted by minimally invasive surgical robots. We introduced the cable transmission mechanism into the design of magnetic surgical instruments. The prototype magnetic surgical forceps we developed feature a 10-mm-diameter manipulator, comparable in size to conventional minimally invasive surgical robots (a typical minimally invasive surgical robot—da Vinci SP—has manipulators with a diameter of 8 mm), which is smaller than the developed magnetic surgical instruments (not including magnetic endoscopes). In the design, the difficulty in connecting and pre-tensioning the driving cables within the compact body of the magnetic surgical instrument was resolved through a pair of pre-tensioning buckles. The decoupling was realized by driving the motors corresponding to the coupled joints to compensate for the length changes of the driving cables. The effectiveness of the decoupling method was tested. The results showed that the angle variations of the coupled joints were within 2°. The practicality of the manipulator has also been verified through a grasping experiment.

In this paper, the design and decoupling control of the manipulator of the magnetic and cable-driven surgical forceps are described. Regarding future works, it is also needed to fully verify the performances of the magnetic and cable-driven surgical instruments: for example, measuring the mechanical properties, conducting ex vivo or in vivo experiments to verify the working ability, and proving the safety.

## Figures and Tables

**Figure 1 micromachines-16-00650-f001:**
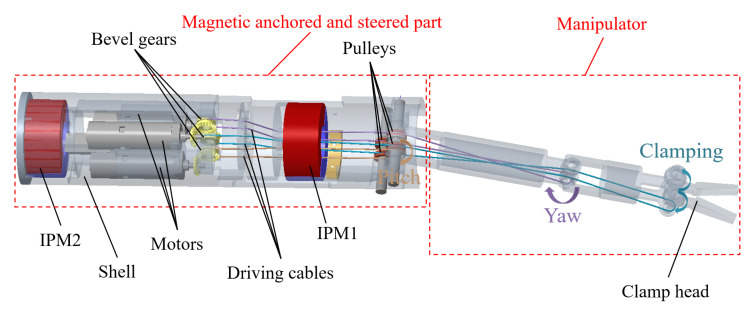
Structural diagram of magnetic surgical forceps.

**Figure 2 micromachines-16-00650-f002:**
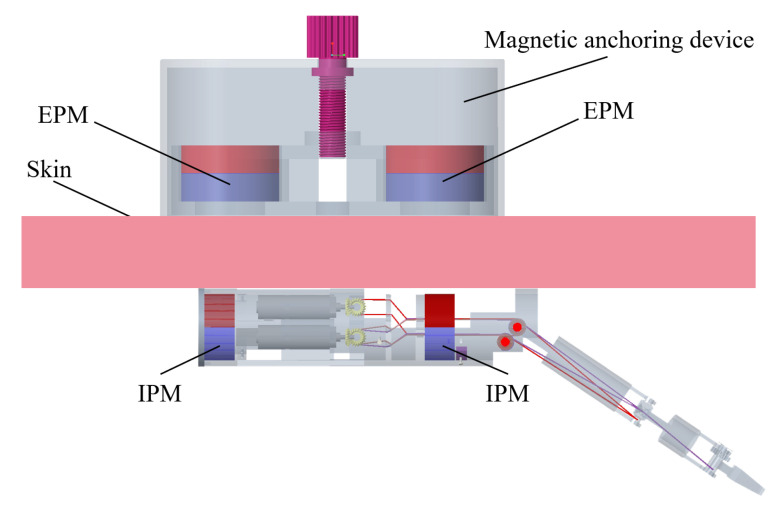
The magnetic surgical forceps are attached to the inner wall of the abdominal cavity by a magnetic anchoring device.

**Figure 3 micromachines-16-00650-f003:**
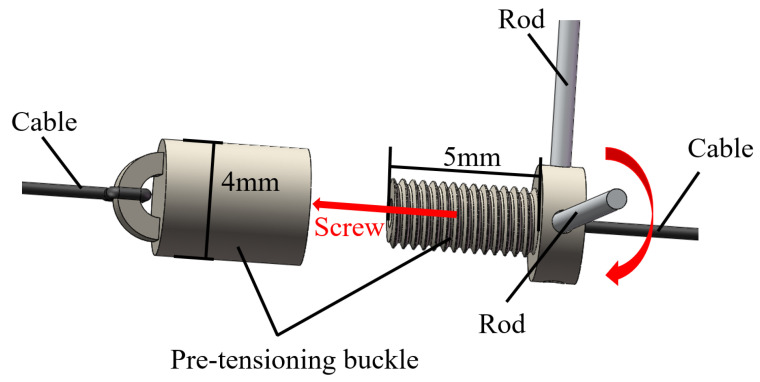
Pre -tensioning buckles. Red curved arrow represents the rotation direction, and the red straight arrow represents the moving direction.

**Figure 4 micromachines-16-00650-f004:**
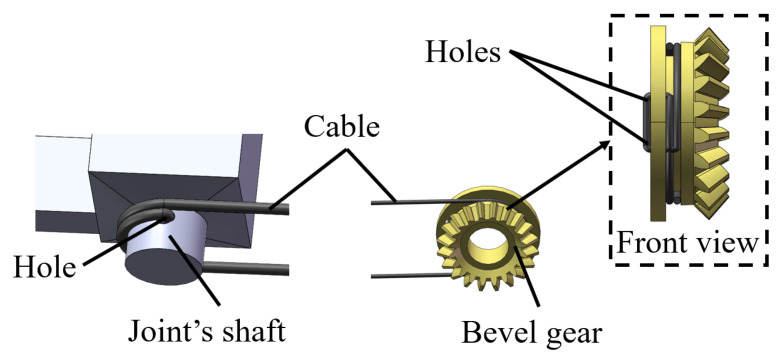
Anti -sliding design.

**Figure 5 micromachines-16-00650-f005:**
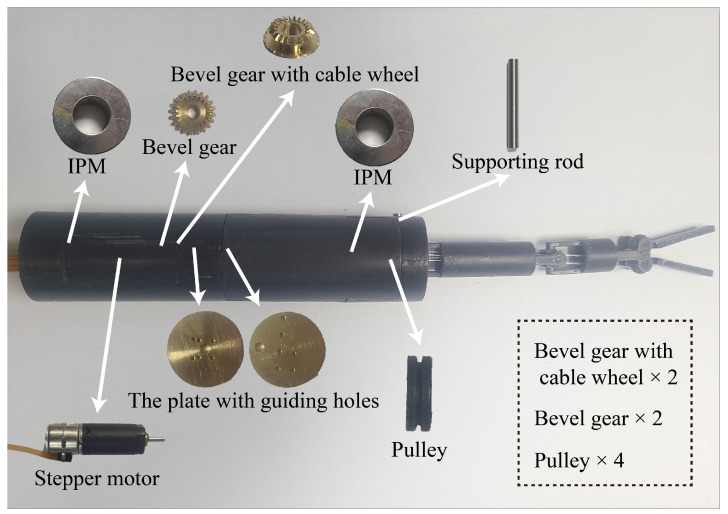
The magnetic anchored and cable-driven surgical forceps prototype and its components. For clearly demonstrating the structure of the parts, some small parts have been enlarged. The bevel gear and the bevel gear with cable wheel are enlarged by a factor of 2, and the pulley is enlarged by a factor of 4.

**Figure 6 micromachines-16-00650-f006:**
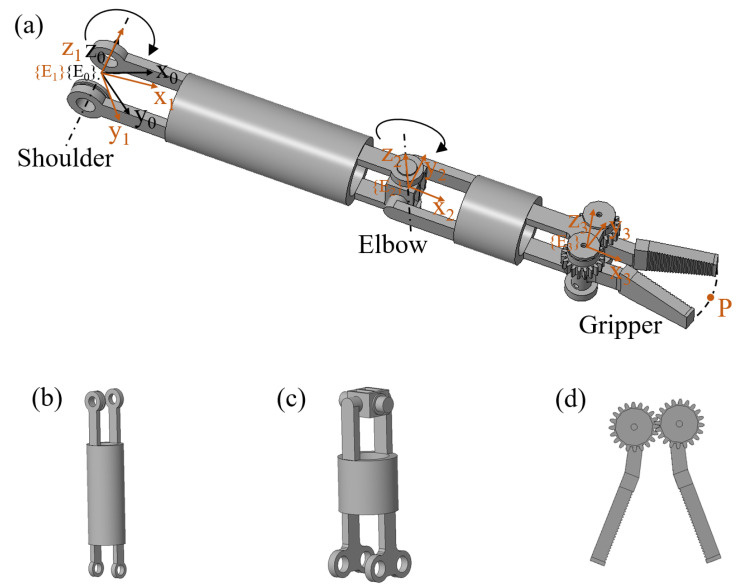
Structure of the manipulator. (**a**) The manipulator and the coordinate systems distribution. (**b**) The pitch joint. (**c**) The yaw joint. (**d**) The clamping joint.

**Figure 7 micromachines-16-00650-f007:**
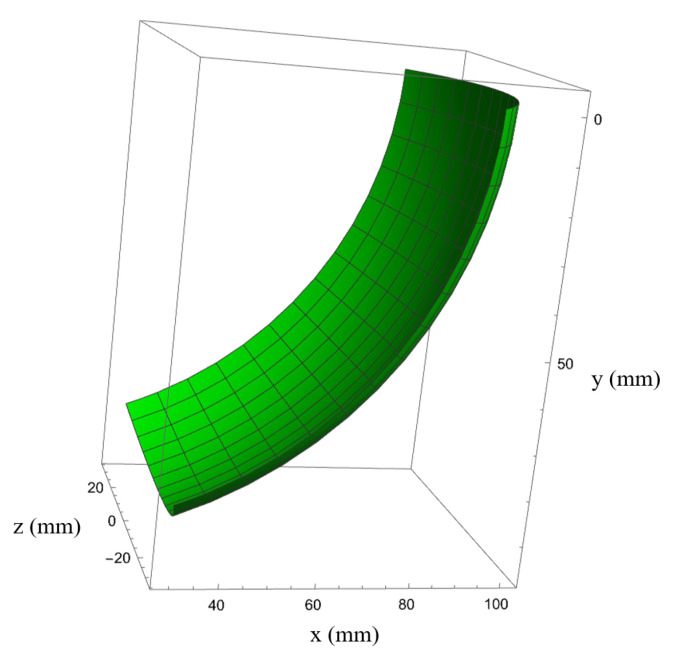
Workspace of the manipulator.

**Figure 8 micromachines-16-00650-f008:**
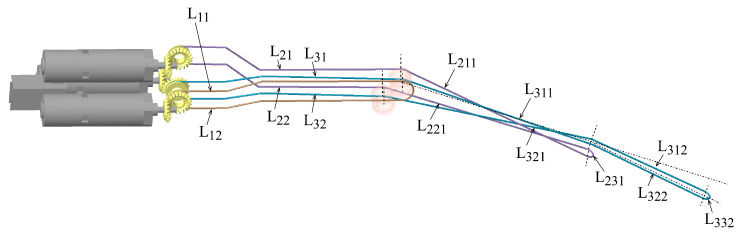
Distribution of the driving cables.

**Figure 9 micromachines-16-00650-f009:**
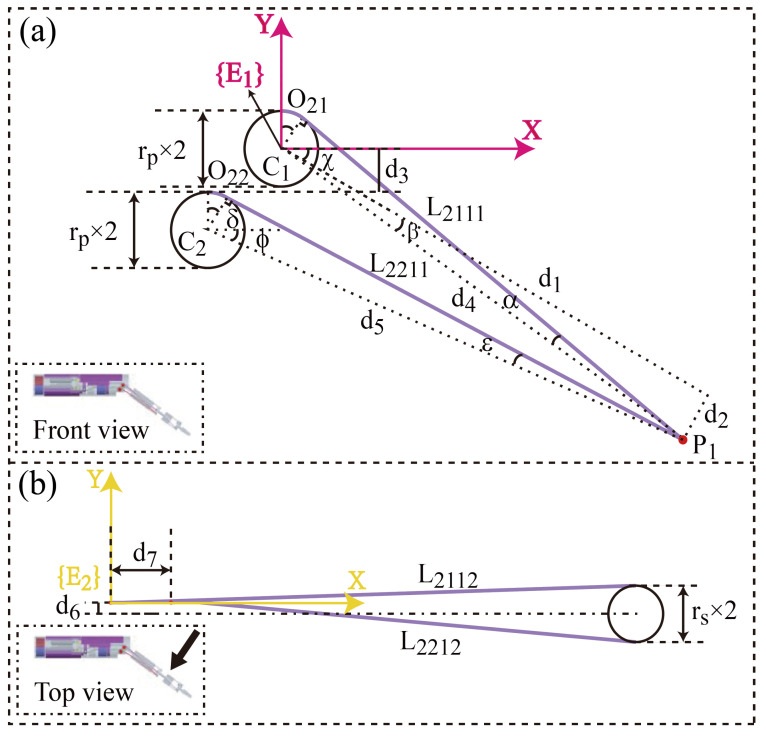
Line diagram of the yaw joint-driving cable sections that pass through the manipulator. (**a**) Front view. (**b**) Top view.

**Figure 10 micromachines-16-00650-f010:**
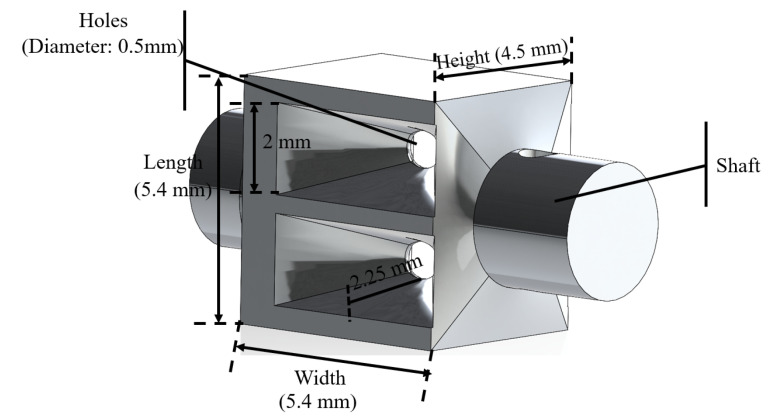
Structure and geometric dimension of the holes for passing through the clamping joint-driving cable.

**Figure 11 micromachines-16-00650-f011:**
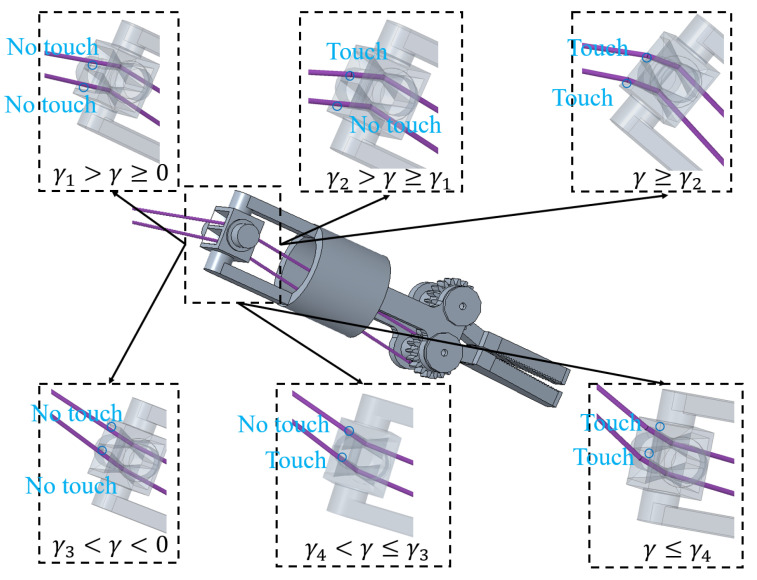
Contact states between the clamping joint-driving cable and the holes in the yaw joint.

**Figure 12 micromachines-16-00650-f012:**
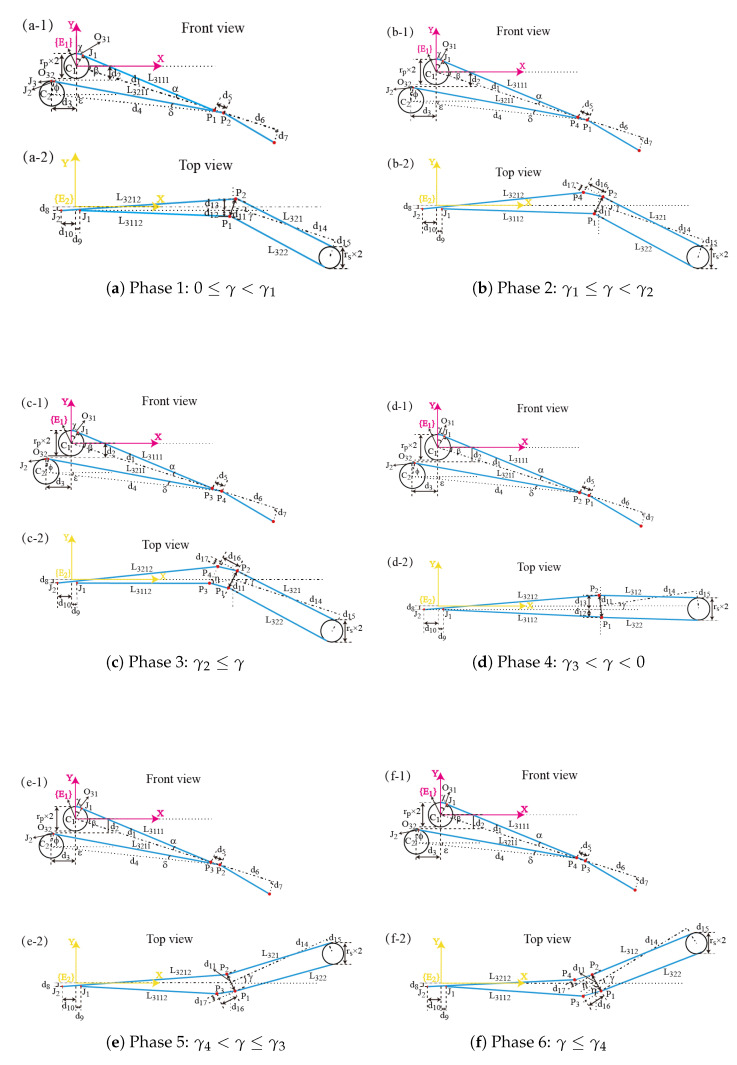
Schematic diagram of the clamping joint-driving cable sections that pass through the manipulator. ((**a**–**c**) The deflection angle of the yaw joint is larger than zero. (**d**,**e**) The deflection angle of the yaw joint is smaller than zero.) (**a**) Neither $L_{311}$ nor $L_{321}$ contacts the sidewalls of the holes. (**b**) $L_{311}$ contacts one sidewall of its respective hole. (**c**) Both $L_{311}$ and $L_{321}$ contact the sidewalls of their respective holes. (**d**) Neither $L_{311}$ nor $L_{321}$ contacts the sidewalls of the holes. (**e**) $L_{321}$ contacts one sidewall of its respective hole. (**f**) Both $L_{311}$ and $L_{321}$ contact the sidewalls of their respective holes.

**Figure 13 micromachines-16-00650-f013:**
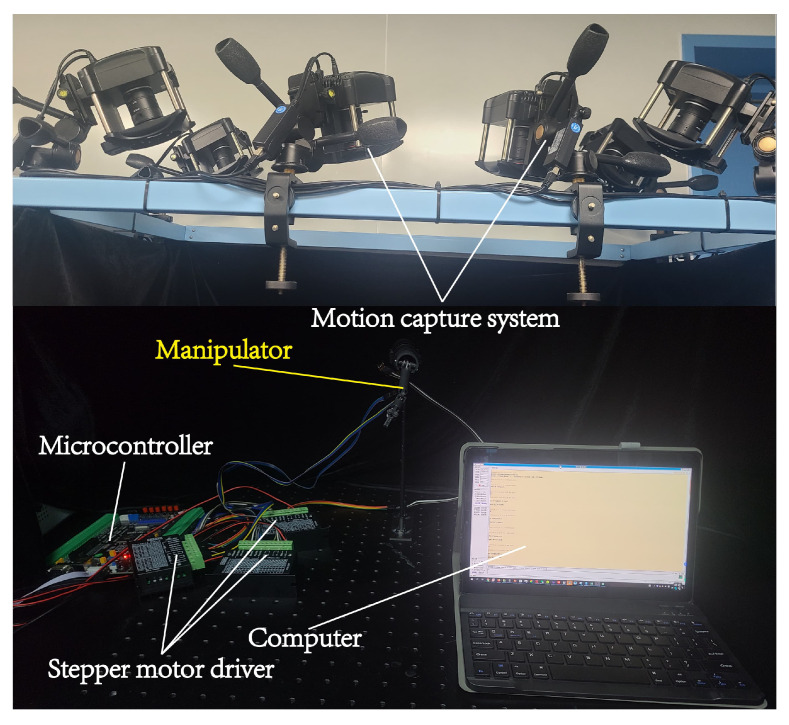
Experimental setup for capturing the motion of the manipulator.

**Figure 14 micromachines-16-00650-f014:**
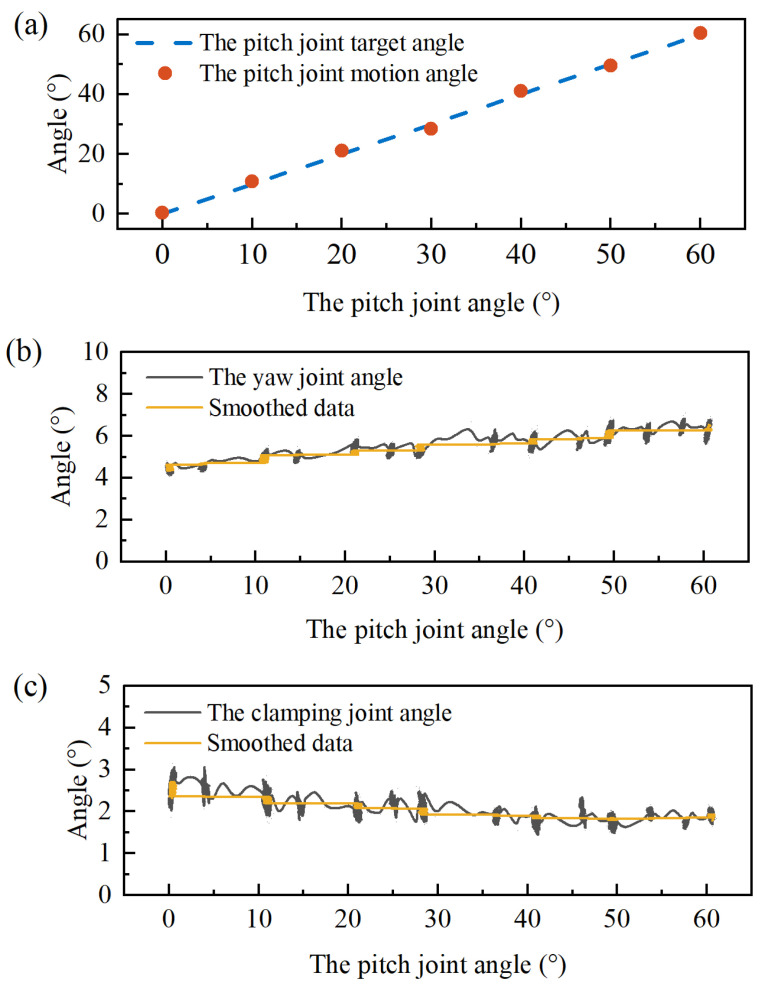
Decoupling effect of the pitch joint. (**a**) The pitch joint rotation from 0° to 60° at intervals of 10°. (**b**) The angle variation of the yaw joint. (**c**) The angle variation of the clamping joint.

**Figure 15 micromachines-16-00650-f015:**
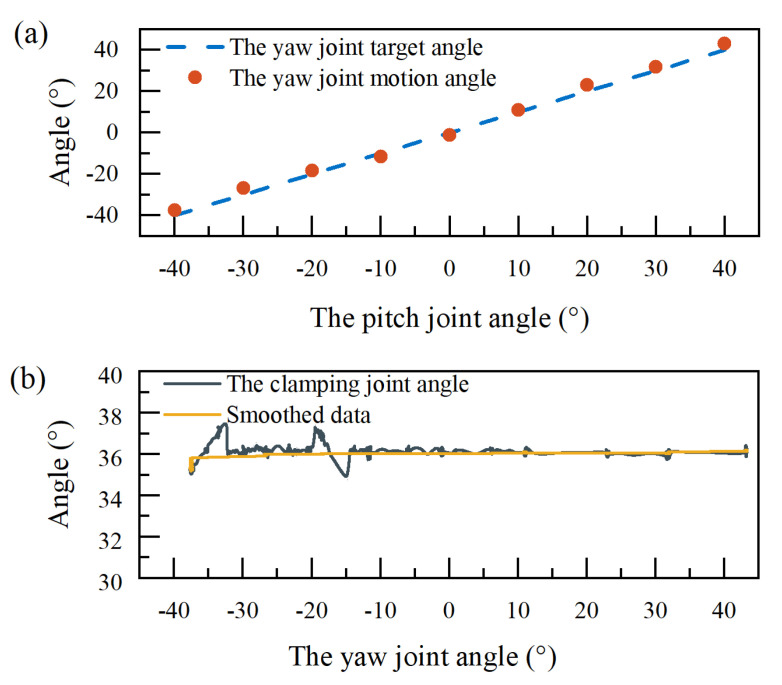
Decoupling effect of the yaw joint. (**a**) The yaw joint rotation from −40° to 40° at intervals of 10°. (**b**) The angle variation of the clamping joint.

**Figure 16 micromachines-16-00650-f016:**
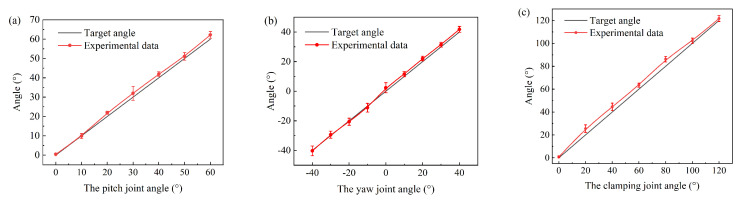
The joint motions of the manipulator. (**a**) The motion of the pitch joint. (**b**) The motion of the yaw joint. (**c**) The motion of the clamping joint.

**Figure 17 micromachines-16-00650-f017:**
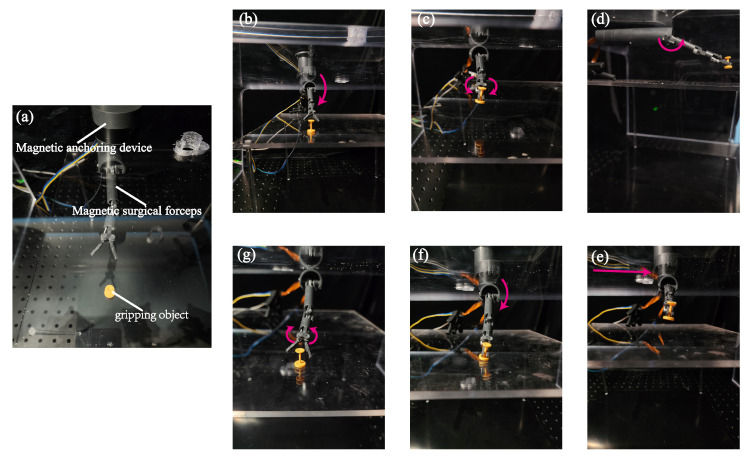
Grasping experiment. (**a**) Experimental setup; (**b**) Approach the gripping object; (**c**) Clip; (**d**) Pick up; (**e**) Translate; (**f**) Put down; (**g**) Release.

**Table 1 micromachines-16-00650-t001:** Parameters of Figure 9.

Symbol	Value (mm)
$d_{1}$	50
$d_{2}$	3.2
$d_{3}$	2.53
$d_{6}$	0.75
$d_{7}$	4
$r_{p}$	2.35
$r_{s}$	1.6

**Table 2 micromachines-16-00650-t002:** Parameters of Figure 12.

Symbol	Value (mm)
$r_{p}$	2.35
$r_{s}$	1.6
$d_{1}$	50
$d_{2}$	2.53
$d_{3}$	4
$d_{4}$	0.75
$d_{7}$	3
$d_{11}$	2.4
$d_{14}$	30
$d_{15}$	3.2
$d_{16}$	2.25
$d_{17}$	1

## Data Availability

No new data were created or analyzed in this study.

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
