# Peer review of "Design and Control of the Manipulator of Magnetic Surgical Forceps with Cable Transmission"

_micromachines, 2025, doi:10.3390/mi16060650_

Round 1
Reviewer 1 Report
Comments and Suggestions for Authors
This paper presents the design and control of a magnetic and cable-driven surgical forceps for minimally invasive surgery. The forceps use cable transmission to separate the power source from the manipulator, significantly reducing its diameter to 10 mm. A novel pre-tensioning buckle design addresses the challenge of connecting and pre-tensioning cables within the compact instrument body. The paper proposes a mathematical model to characterize the length changes of coupled joint driving cables and integrates it into the control program to achieve joint motion decoupling. Experiments validate the decoupling effect, with coupled joint angle variations kept within 2°. The study demonstrates the feasibility of using cable transmission in magnetic surgical instruments and lays the foundation for further development of compact and dexterous surgical robots. This paper is well-structured and well-written. Before publication, there are some questions to be solved.
- The authors mentioned “Compared with open surgery, minimally invasive surgery (MIS) significantly reduces surgical trauma to the human body with a better recovery effect.”, more state-of-the-art can be cited: DOI: 10.34133/cbsystems.0083; DOI: 10.34133/cbsystems.0188; DOI: 10.34133/cbsystems.0110.
- How to ensure consistent tension across all cables based on the pre-tensioning buckle design, and how to achieve uniform tension.
- How to achieve decoupling of joint motions, and how to compensate for the length changes in real-time.
- The authors mentioned that the angle variations of the coupled joints were within 2°, how about the average and maximum angle variations during the experiments.
- How to quantify the accuracy of the manipulator evaluated.
- The scale bars of the figure 11 and 15 are missing, which would help identify the size.
Reviewer 2 Report
Comments and Suggestions for Authors
- Figure 11. Experimental setup for capturing the motion of the manipulator. The English ‘Compulater’ in the figure should be ‘Computer’. Please double-check the English presentation in the paper.
- It is suggested to add test results under different loads in the conclusion part of the paper, which is important for real surgical scenarios.
- Experimental validation is limited to idealized conditions. Comparative analysis with state-of-the-art magnetic surgical instruments (e.g., diameter, payload, degrees of freedom) and testing in a biomimetic environment would better highlight the advantages and practical limitations of the design.
- Does this paper consider stability under external perturbations?
Reviewer 3 Report
Comments and Suggestions for Authors
The paper is well-written in general. Here are some comments to improve the quality of the paper.
- The effect of friction between tendons and pulleys should be discussed in details.
- A close-up picture of prototype should be provided showing different elements.
- It is not clear how smooth is the motion of the gripper.
- the effect of backlash/hysteresis should be explained.
- It is not clear how the tool can be actuated by magnets. Stepper motors were used.
Round 2
Reviewer 1 Report
Comments and Suggestions for Authors
Publish
Reviewer 2 Report
Comments and Suggestions for Authors
The review comments have been responded to, the quality of the paper has been greatly improved, and the paper is ready for publication.